# Gastrointestinal Involvement in Primary Antibody Deficiencies

**Tomas Milota [1],\*** , **Jitka Smetanova [1]** and **Iveta Klojdova [2]**

1 Department of Immunology, Second Faculty of Medicine Charles University and Motol University Hospital, 15006 Prague, Czech Republic
2 DRIFT-FOOD, Faculty of Agrobiology, Food and Natural Resources, Czech University of Life Sciences, 15006 Prague, Czech Republic
\* Correspondence: tomas.milota@fnmotol.cz; Tel.: +420-22-443-5961

**Abstract:** Primary antibody deficiencies (PADs) are the most frequent group of inborn errors of immunity. Impaired B-cell development, reduced production of immunoglobulins (mainly IgG and IgA), and specific antibodies resulting in recurrent infections are their hallmarks. Infections typically affect the respiratory tract; however, gastrointestinal involvement is also common. These include infection with *Helicobacter pylori*, *Salmonella*, *Campylobacter* species, *Giardia*, and noroviruses. Impaired IgA production also contributes to dysbiosis and thereby an increase in abundance of species with proinflammatory properties, resulting in immune system dysregulation. Dysregulation of the immune system results in a broad spectrum of non-infectious manifestations, including autoimmune, lymphoproliferative, and granulomatous complications. Additionally, it increases the risk of malignancy, which may be present in more than half of patients with PADs. Higher prevalence is often seen in monogenic causes, and gastrointestinal involvement may clinically mimic various conditions including inflammatory bowel diseases and celiac disease but possess different immunological features and response to standard treatment, which make diagnosis and therapy challenging. The spectrum of malignancies includes gastric cancer and lymphoma. Thus, non-infectious manifestations significantly affect mortality and morbidity. In this overview, we provide a comprehensive insight into the epidemiology, genetic background, pathophysiology, and clinical manifestations of infectious and non-infectious complications.

**Keywords:** selective immunoglobulin A deficiency; common variable immunodeficiency; X-linked agammaglobulinemia; gastrointestinal tract; microbiome; celiac disease; inflammatory bowel disease; gastric cancer; lymphoma

## 1. Introduction

Primary antibody deficiencies (PADs) are the most prevalent inborn errors of immunity (IEI) [1]. The clinical spectrum ranges from the selective immunoglobulin (Ig) class (such as selective IgA deficiency (sIgAD)) and the IgG (IgG1-IgG4) subclass deficiencies to severe disorders of antibody production such as X-linked agammaglobulinemia (XLA) or common variable immunodeficiency (CVID) [2]. Recurrent bacterial respiratory tract infections are a hallmark of PADs, but severe gastrointestinal tract (GIT) infections are frequent [3–5]. Patients with PADs may also be affected by a broad spectrum of non-infectious complications that may significantly contribute to morbidity and mortality, which develop based on immune dysregulation. The non-infectious manifestations include autoimmune, lymphoproliferative, and granulomatous diseases, which cause autoimmune cytopenia, splenomegaly, lymphadenopathy, nodular lymphoid hyperplasia, granulomatous lymphocytic interstitial lung disease, and a broad spectrum of PAD-associated gastropathies and enteropathies [6–8]. There is also an increased risk of hematological, lymphoproliferative, and solid organ malignancies [9]. Therefore, early and appropriate diagnosis and treatment of PADs and the related complications play an important role in prognosis [10,11] and may have a significant impact on quality of life [12,13]. Additionally, owing to the

main features mimicking various autoimmune and inflammatory diseases, such as celiac disease (CED) and inflammatory bowel disease (IBD), gastrointestinal (GI) involvement is one of the most challenging complications of PADs. In this overview, we discuss the spectrum of GI disorders affecting patients with PADs (Table 1), which include mechanisms of immune system dysregulation, genetic background, the role of the microbiome in non-infectious complications, and diagnostic and therapeutic approaches. We primarily focused on sIgAD, CVID, and inherited agammaglobulinemia as the most prevalent PADs. The review was prepared in line with the Preferred Reporting Items for Systematic reviews and Meta-Analyses guidelines [14] and the proposed guidelines for biomedical narrative review preparation [15].

**Table 1.** Summary of the most common infectious, non-infectious and malignant complications associated with Primary antibody deficiencies.

| **Infections** | |
| --- | --- |
| Helicobacter pylori | *Campylobacter* spp. |
| *Salmonella* spp. | Gardia lamblia |
| Noroviruses | Enteroviruses |
| **Non-infectious manifestation** | |
| Chronic gastritis (gastropathy) | Pernicious anemia |
| Celiac (-like) disease | Inflammatory bowel (-like) disease |
| Nodular lymphoid hyperplasia | |
| **Malignancy** | |
| Gastric metaplasia (precancerous lesion) | Gastric cancer |
| Lymphoma | |

## 2. Epidemiology

PADs represent more than half of IEI. sIgAD is the most common PAD, with incidence ranging from 1:1000 to 1:140. Their incidence is higher in Caucasians but lower in Asian countries [16,17]. CIV is the second most common PAD, with a prevalence ranging from 0.08 (Poland) to 3.14:100,000 (Denmark) in Europe and 1.48 in USA [18]. However, other PADs are rare. The reported prevalence of XLA is 1–2:100,000, accounting for approximately 85% of all cases. However, autosomal dominant and recessive forms have been described as affecting the genes involved in lymphopoiesis and B-cell receptor development [19]. Hypogammaglobinemia may also accompany other types of IEI, such as combined immunodeficiencies and diseases of immune dysregulation [2].

### 2.1. Epidemiology of Selective IgA Deficiency (sIgAD)

Although the majority of patients with sIgAD are asymptomatic, infectious manifestations are found as the first manifestation in 40–90% of symptomatic patients, and the occurrence of autoimmune complications range from 5% to 30%. Patients with a history of autoimmunity were older [20]. Among the sIgAD-associated auto-immunities, CED is of particular interest. sIgAD is 10–15 times more common in patients with CED and affects approximately 2% of patients with CED [21,22]. In contrast, CED was found in 14% of 184 pediatric patients with sIgAD, in contrast to other auto-immunities found in 9% of them. There is also an association between sIgAD and IBD. A population-based study in Sweden reported an overall prevalence of IBD of 3.9%, CD of 2.4%, and UC of 1.7% corresponding to a five times higher prevalence than in the general population. The prevalence ratios of CD and UC were 5.7 and 3.9, respectively. However, a prevalence of 6.5% and a ratio of 35 make CED the most frequent autoimmune complication in patients with sIgAD [23].

### 2.2. Epidemiology of Common Variable Immunodeficiency (CVID)

Non-infectious inflammatory conditions can affect up to half of the patients with CVID. Autoimmune diseases may affect approximately 30% of all patients and represent

the most common non-infectious complication, followed by chronic lung disease and lymphoproliferative disorders. GI involvement has been reported in more than 15% of cases; it includes a broad spectrum of disorders. In a US cohort of 623 patients, the reported prevalence of GI diseases was 17.3% [7]. However, in a Finnish study with 132 patients, endoscopic or histological abnormalities were observed in 58% and 68% of patients who underwent upper and lower GIT endoscopy, respectively [24]. A recent meta-analysis showed that the spectrum of the most common manifestations of GIT disorders in CVID include chronic diarrhea (27%, 95% CI: 21–34), gastritis (28%, 95% CI: 22–35), gastric metaplasia (25%), gastroesophageal reflux disease (16%, 95% CI:6–25), malabsorption (13%, 95% CI: −4–26), villous atrophy (11%, 95% CI: 1–28), and IBD (10%, 95% CI: 6–16) [25].

### 2.3. Epidemiology of X-Linked Agammaglobulinemia (XLA)

Infections are present in the majority of XLA cases (up to 85%) [26,27]. Gastroenteritis, particularly bacterial ethology, represents the most common manifestation of gastrointestinal involvement [28]. However, patients with XLA seem to be prone to the enteroviruses causing a variety of symptoms including fever, headache, respiratory illness, sore throat, or vomiting and diarrhea [5]. Several members of the enterovirus family may also affect the nervous system [29]. Nervous system involvement within enteroviral infections is associated with high mortality in these patients [30]. Unusual complications, such as IBD and large granular lymphocyte disease, were observed in 20.3% of 783 patients reported in a multicenter study [31]. Another study including 128 patients found inflammatory symptoms in 69% of the patients; however, only 28% were diagnosed with any inflammatory condition. Similarly, in another report, 21% of the patients reported chronic diarrhea and 17% had abdominal pain; however, Crohn's disease (CD) was diagnosed in only 4% of the patients [32].

## 3. Genetic Background

Generally, PADs are regarded as multifactorial polygenic diseases in most sporadic cases, with the exception of well-defined syndromes, such as inherited agammaglobulinemia or hyperIgM syndromes. To date, 45 different genes have been identified as monogenic causes of PADs [33]. The majority of these genes drive B-cell development, and their mutations lead to a developmental blockage. However, a number of genes also affect T-cell function, resulting in impaired function.

### 3.1. Monogenic Causes of sIgAD

Mutations in *JAK3, RAG1, DCLRE1C, CD27, LRBA, BTK, TACI, TWEAK, MSH6, MSH2, PIK3R1,* and *CARD11* were associated with sIgAD development [34]. There was also an association with certain human leukocyte antigens: HLA-A1, HLA-B8, HLA-DR3, and HLA-DQ2 [11]. In particular, HLA-DQ2 is strongly associated with CED; HLA-DQ2 contains immunogenic gluten peptides and triggers an immune response [35]. A similar spectrum of gene defects has been described in cases of monogenic CVID. Several reports have described the progression from sIgAD to CVID, suggesting a close association between the two diseases [36,37].

### 3.2. Monogenic Causes of CVID

Mutations in *PIK3CD, PIK3R1,* nuclear factor kappa-light-chain-enhancer of activated B cells 1/2 (*NFkB1/2*), cytotoxic T-lymphocyte antigen 4 (*CTLA4*), and *LRBA* are the most clinically relevant [38–40]. Although monogenic causes of CVID have been identified in approximately 10% of patients, genetic testing plays an important role in many aspects of patient care. Genetic testing allows us to make a definitive diagnosis, assess prognosis, identify patients for specific therapies, and support family planning decisions [41]. Gain-of-function mutations in *PIK3CD* and loss-of-function mutations in *PIK3R1* are associated with activated PI3K-delta syndrome (APDS) types 1 and 2, respectively. PI3 kinases (PI3Ks) are a family of lipid kinase enzymes producing 3′-phosphorylated phosphoinositides. They are

activated by the engagement of various receptors, including BCR and TCR. These lipids act as secondary messenger molecules. Two key pathways resulting from the action of PI3Ks are the activation of NF-κB through the PLCγ-DAG/IP3-PKC pathway and Akt-mTOR. These pathways particularly control the production of proinflammatory cytokines and cell metabolism [42,43].

CTLA4, a CD28 homologous glycoprotein, is constitutively expressed on Tregs and plays an essential role in their function. CTLA4 competitively binds to CD80 and CD86 (B7 proteins); however, it possesses the opposite effect compared with the activation of CD28, and the binding activity is expected to be 10–100 times. As a result, T cells receive fewer CD28-mediated activation signals. Moreover, CTLA4 induces the removal of B7 proteins from the cell surface via transendocytosis. The intracellular trafficking of CTLA4 is controlled and regulated by specific proteins, such as clathrin adaptor complexes. On the other hand, the turnover of CTLA4 is promoted by the lipopolysaccharide (LPS)-response and beige-like anchors (LRBA), a protein protecting CTLA4 from lysosomal degradation. This mechanism facilitates accumulation of CTLA4 molecules in the cytoplasm, followed by subsequent cell membrane re-expression. CTLA4 and LRBA play critical roles in peripheral tolerance mediated by Tregs. Their impaired function may lead to variable phenotypes that are usually characterized by autoimmune phenomena; hypogammaglobulinemia; and decreased Treg, class-switched (cs)-B-cell, plasmablast, and follicular T-helper-cell count [44,45].

### 3.3. Genetic Background of XLA

Mutations in the X-linked *BTK* gene encoding Bruton's tyrosine kinase are the most common causes of inherited agammaglobulinemia. To date, more than 600 mutations have been described, 10–15% of which occur de novo. Most of them involve one to four base pairs. However, larger deletions have been found in 3–5% of patients. The deleted regions may damage other linked or adjacent genes such as TIMM8A and TAF7L, resulting in both XLA and deafness-dystonia-optic neuropathy syndrome. BTK is a cytoplasmic tyrosine kinase expressed mainly on hematopoietic cells. It is essential for BCR-mediated proliferation and survival and B-cell development. BTK transduces signals from pre-BCR complexes comprising μ heavy chain and surrogate light chain proteins (VpreB and λ5), and Ig-α/CD79a and Ig-β/CD79b signaling components. Impaired BTK function leads to the developmental blockage of pre-B cells in the bone marrow and missing B cells in the peripheral blood. Nevertheless, hypomorphic mutations may cause a partial interruption, leading to a "leaky" phenotype. Mutated BTk may also be associated with neutropenia in some patients. Notably, patients with a known family history of XLA had an earlier age of diagnosis than patients without a family history (mean age 2.59 vs. 5.37 years). Autosomal dominant (LRRC8, TOP2B) and recessive forms (μ heavy chain, Lambda 5, Ig alpha/beta, BLNK, PI3K genes, and TCF3) represent rare forms of inherited agammaglobulinemia. Most of them affect pre-B-cell/B-cell receptor complex or signaling pathways [19,46–49].

### 4. Mechanisms of Immune Dysregulation

Immune dysregulation is the main feature contributing to the development of inflammatory and autoimmune complications. While the developmental block of B cells in inherited agammaglobulinemia occurs at the pre-B-cell stage, it predominantly affects memory cells in sIgAD and CVID (Figure 1).

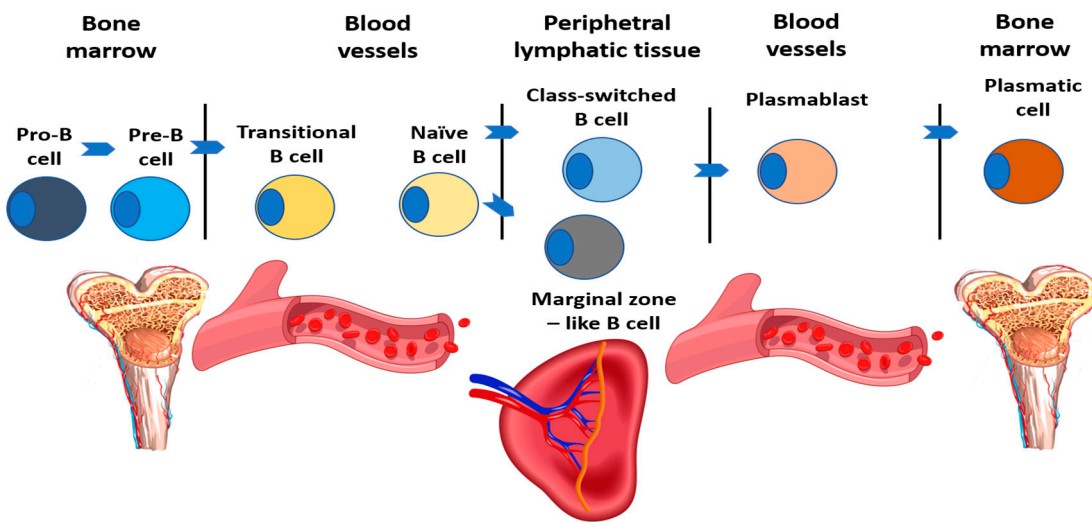

**Figure 1.** B cell development scheme, adapted from Warnatz et al. (2008) [50].

*4.1. Immune System Dysregulation in sIgAD*

Several studies have reported a decreased number of cs-memory B cells, plasmablasts, and transitional B cells in patients with sIgAD. In contrast, the patients had elevated counts of CD21(low)CD38(low) B cells (Table 2) [51].

**Table 2.** Characteristics, relative (% of CD19+ cells), and absolute counts of B-cell subpopulations in peripheral blood [50].

| Subpopulation | Characteristics | Relative Count (%) | Absolute Count (E9/l) |
|---|---|---|---|
| B cells (total) | CD19+ | 6–22 | 0.1–0.53 |
| Transitional B cells | CD19 + IgM + IgD + CD24 + CD38 + CD27- | 0.9–5.7 | 0–0.03 |
| Naïve B cells | CD19 + IgD + CD27- | 48.4–79.7 | 0.06–0.47 |
| Class-switched B cells | CD19 + CD27 + IgD-IgM- | 8.3–27.8 | 0.02–0.09 |
| Plasmablasts | CD19 + CD27 + CD38 + IgM-IgD-CD24- | 0.4–2.4 | 0–0.01 |
| Marginal zone-like B cells | CD19 + IgD + CD27+ | 7–23.8 | 0.01–0.08 |
| CD21(low)CD38(low) B cells | CD19 + CD21lowCD38low | 1.6–10 | 0.01–0.02 |

The B cells of the patients with sIgAD also failed to respond to Toll-like receptor 9 to CpG, which did not induce IgA production and further enhanced transitional B-cell defects. Additionally, it affects the B regulatory subset (Bregs), which express anti-inflammatory interleukin (IL)-10 [52]. Mouse models of IgA deficiency were prone to spontaneously induced inflammation of the ileum [53]. The level of immune dysregulation correlates with the clinical phenotype. Patients with sIgAD with a profound reduction in cs-B-cell count also show more severe clinical features, including pneumonia, bronchiectasis, and autoimmune diseases [54]. Recent findings also implicate the defects of T-helper-cell (Th) subsets, Th1 and Th17, which are reduced along with increased blood concentrations of transforming growth factor (TGF) β1, B-cell activating factor (BAFF), and proliferation-inducing ligand (APRIL) [55]. The BAFF/APRIL system has been identified as an important pathophysiological pathway, and increased levels of both cytokines have been found in a broad spectrum of autoimmune diseases [56]. An increased concentration was also found in patients with CD, and its levels correlated with disease activity [57–59] and even predicted tumor necrosis factor alpha (TNFα) inhibition [60].

*4.2. Immune System Dysregulation in CVID*

A similar spectrum of B-cell changes were observed in patients with CVID. In the majority of patients, the number of cs-B cells was associated with a decrease in serum IgG and IgA levels. In contrast, elevated counts of CD21(low)CD38(low) B cells were

significantly associated with splenomegaly, granulomatous complications, and elevated counts of transitional B cells with lymphadenopathy. CD21(low)CD38(low) B cells showed increased expression of activation markers (such as CD69, CD80, CD86), chemokine receptors recruiting cells to the site of inflammation (e.g., CCR1, CCR5, CCR6) [61]. These cells have features of the competent antigen-presenting cells [62] and the drive inflammatory response towards Th1 [63]. This population also contains mostly autoreactive unresponsive clones [64] and its expansion was found in many autoimmune conditions such as rheumatoid arthritis or systemic lupus erythematodes [65]. Low numbers of memory B cells were also observed in individuals affected by CVID-associated chronic diarrhea [66]. Based on these findings, a few classification systems, such as EUROclass (Figure 2) [67] and Freiburg scheme, have been defined to identify patients at high risk of non-infectious complications [50].

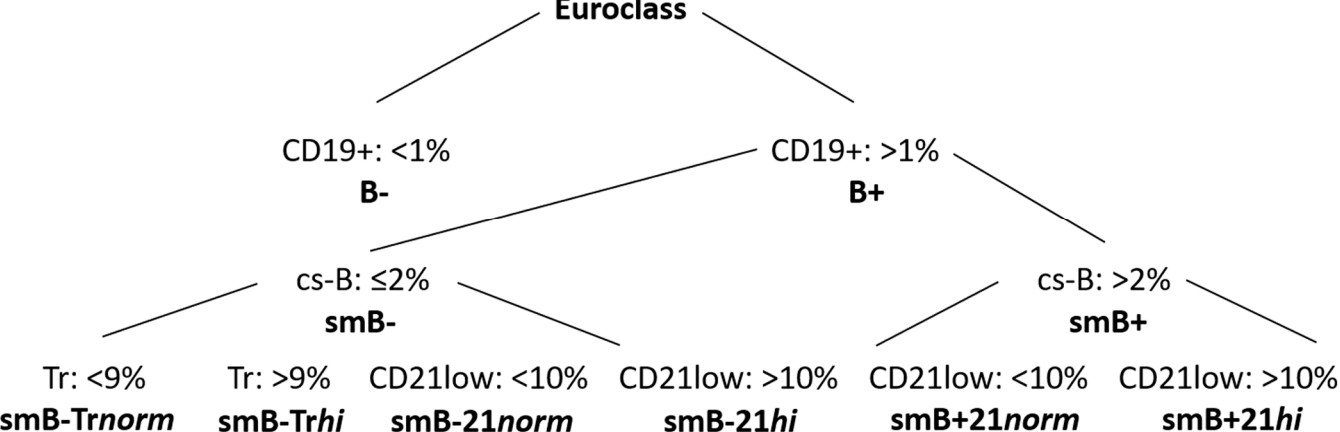

**Figure 2.** Euroclass classification of Common variable immunodeficiency (B, B cells; cs/sm, class-switched/switched-memory B cells; Tr, transitional B cells; norm, normal count; hi, high count) [67].

Reduced Bregs have also been reported in CVID, but no correlation with clinical and immunological characteristics have been found [68]. However, impaired Bregs function may contribute to T-cell activation and immune response skewing toward Th1, characterized by overproduction of IFNγ [69]. Local overproduction of IL-12, a cytokine that drives the immune response toward Th1, was found in patients with CVID-associated IBD. The cytokine profile differentiates between CD and CVID-associated enteropathy, which does not lead to excess levels of IL-23, IL-17, and TNFα [70]. A negative correlation was observed between the frequency of Tregs and the CD21(low)CD38(low) B-cell population [71]. Again, there was no significant correlation between Treg count and clinical autoimmune complications. Nevertheless, the functional defects of Tregs, i.e.,CTLA4 or IL-10, may contribute to immune dysregulation as CTLA4 deficiency has been described as one of the monogenic causes of CVID associated with severe autoimmune complications [72]. Heavily impaired production of IL-10 or function of the IL-10 receptor is closely related to severe early-onset forms of IBD [73]. Other T-cell abnormalities include increased T-cell activation, apoptosis, and exhaustion, which may result in reduced T-cell counts [74,75]. Severely reduced CD4+ T cells (<200 × $10^6$ cells/L) were found in specific subgroups of CVID and late-onset combined immunodeficiency (LOCID). These patients had a higher frequency of splenomegaly, granulomas, GI diseases, and lymphomas [76]. LOCID is more common in patients with monogenic CVID [77].

## 5. GI Infections

Recurrent sinopulmonary infection is a hallmark of PADs. Clinically, sIgAD is characterized by undetectable serum levels of IgA as the main diagnostic criterium. Although IgA plays an important role in mucosal immunity, most patients are asymptomatic. More severe infections and post-infectious complications, such as bronchiectasis, may be associated



with IgG2 deficiency. Patients with PADs are also predisposed to develop GI infections. Although *Helicobacter pylori* (HP) infection was found to be one of the most common causes of GI infections in pediatric patients with sIgAD [78], no differences in proportion were observed between a cohort of older patients and in the general population. However, HP infection is associated with more severe forms of gastritis, duodenal ulcers, nodular lymphoid hyperplasia, and gastric cancers [79,80]. Similarly, CVID does not predispose to HP infection, but is associated with immune dysregulation, including expansion of CD21(low)CD38(low) B cells or reduced expression of CD25 on T cells, similar to sIgAD. In the context of other findings, such as atrophic gastritis and metaplasia, HP infection may contribute to the development of gastric cancer, one of the most common CVID-associated malignancies [81]. Early HP screening and eradication, upper endoscopy, and assessment of other risk factors such as increased serum vitamin B12 levels and iron levels should be actively offered to all patients with CVID [82]. Another GI infection described in patients with PAD is Giardia lamblia [83,84]. Although T-cell immunity is important for the clearance of Giardia infection [85], impaired barrier function allows its adherence to the epithelium and subsequent proliferation and infection development. Giardia infection is diagnosed based on the clinical manifestations of bloating, cramping, or watery diarrhea, and the results of stool examination. In cases of negative results and persistent suspicion of Giardia infection, duodenal aspirates should be indicated as an alternative test with a higher sensitivity. Metronidazole is the treatment of choice, but infection is often unremitting in sIgAD [86]. The spectrum of pathogens also include *Salmonella* spp. and *Campylobacter* spp. [87]. Despite a predisposition to bacterial infections, there is also an increased risk of viral infection. Viral positivity is associated with mucosal inflammation, and reduced serum and secretory IgA levels have been shown to be significant predispositions [88]. Notably, norovirus has been identified as a cause of CVID-associated enteropathy including intestinal villous atrophy and malabsorption [89]. The clearance of norovirus leads to symptom resolution and histological recovery, induced after ribavirin therapy in some patients [90]. Immunoglobulin replacement therapy (IRT) is the key therapeutic option in the management of infectious complications in PADs. IRT with intravenous or subcutaneous immunoglobulins is indicated in severe hypogammaglobulinemia, disturbed specific antibody immune response, and severe recurrent infections with or without concomitant antibiotic therapy [91].

## 6. The Role of Microbiome

IgA also contributes to the control of gut microbiota composition that cannot be substituted by IgM, which shows less specificity [92]. Insufficient production of IgA leads to dysbiosis, and the microbiota of patients with sIgAD is enriched with species with proinflammatory properties [93]. Furthermore, gut dysbiosis is another factor contributing to the pathogenesis of enteropathy in CVID. The findings from several studies suggest a role for altered microbiota in systemic immune response activation [94,95]. For instance, pathobionts such as Acinetobacter baumannii may re-direct pathways from lipid metabolism to immune response related to enteropathies [96]. In contrast, *Bifidobacterium*, commonly reduced in CVID, ameliorated the gut barrier, and reduced systemic inflammation in mouse models. Interestingly, there was a limited impact on the microbiome with the repeated antibiotic therapies that are frequently used in CVID [97].

## 7. Non-Infectious Manifestation

The main feature of PADs is the impaired production of immunoglobulins. sIgAD is characterized by undetectable serum IgA levels, and CVID is defined by significantly reduced IgA and IgG levels (<2 SD of the normal levels) with variable levels of IgM that are missing in XLA along with all remaining immunoglobulin classes. CVID is accompanied by a disturbed specific response to polysaccharide and/or protein antigens, which should be preserved in sIgAD. Heavily reduced counts of B cells in the peripheral blood (<2%) are found in XLA. XLA is typically diagnosed before 5 years of age, and a definitive diagnosis of sIgAD and CVID should be made after 4 years of age. Secondary causes of

antibody deficiency should be considered and excluded. Autoimmune, lymphoproliferative, and granulomatous complications represent other features that are reflected in the diagnostic criteria of sIgAD and CVID, in addition to recurrent infections [98]. However, non-infectious complications may be diagnosed in up to 28% of patients with XLA, as inflammation-related symptoms are reported more often in a majority of patients [32]. The diagnostic criteria are summarized in Table 3.

**Table 3.** Diagnostic criteria of European Society for Immunodeficiency (ESID) for agammaglobulinemia, common variable immunodeficiency, selective IgA deficiency [98].

| **Agammaglobulinemia** |
| --- |
| <2% of circulating B cells |
| normal number of T cells |
| <200 mg/dL in infants aged < 12 months or |
| <500 mg/dL in children aged > 12 months |
| or normal IgG levels with IgA and IgM below 2SD |
| onset of recurrent infections before 5 years of age |
| **Common variable immunodeficiency** |
| increased susceptibility to infection |
| autoimmune, granulomatous, lymphoproliferative manifestations |
| affected family member with antibody deficiency |
| marked decrease in IgG and marked decrease in IgA with or without low |
| IgM levels (<2SD for specific age) |
| poor antibody response to vaccines and/or absent isohemagglutinins |
| secondary causes of hypogammaglobulinemia have been excluded |
| diagnosis is established after the 4th year of life |
| no evidence of profound T-cell deficiency |
| **Selective IgA deficiency** |
| increased susceptibility to infection |
| autoimmune manifestations |
| affected family member |
| undetectable serum IgA (<0.07 g/L) but normal serum IgG and IgM |
| normal IgG antibody response to all vaccinations |
| diagnosis after 4th year of life |
| secondary causes of hypogammaglobulinemia have been excluded |
| exclusion of T-cell defect |

### 7.1. Non-Infectious Complications of sIgAD

Autoimmune conditions associated with sIgADs include idiopathic thrombocytopenic purpura, Graves' disease, autoimmune hemolytic anemia, type 1 diabetes mellitus, rheumatoid arthritis, thyroiditis, and systemic lupus erythematosus [11,99]. Despite the absence of anti-gliadin, endomysium, and transglutaminase IgA autoantibodies, patients with sIgAD may develop villous atrophy and permanent gluten intolerance. The diagnosis of CED in a terrain of absent IgA is very challenging. IgG class CED-specific antibody assessment and/or endoscopy with histological verification should be considered. Specific IgG antibodies can be used to monitor a patient's dietary compliance. It is also recommended to assess the total serum IgA concentration when patients are tested for CED and the presence of specific IgA antibodies [22,100]. Moreover, the threshold also indicates the strong association of CED with type 1 diabetes mellitus, the second most common autoimmune complication in sIgAD [101]. Interestingly, ulcerative colitis [102], CD [103–105], and nodular lyphoid hyperplasia [106,107] have been reported less frequently in sIgAD than in CVID.

### 7.2. Non-Infectious Complications of CVID

In a cohort of 473 patients with CVID who were followed up for 4 decades, approximately 70% of the patients experienced one or more inflammatory autoimmune manifestations. Chronic lung disease and hematologic and organ-specific autoimmune diseases (immune thrombocytopenic purpura and autoimmune hemolytic anemia as the

most common) were reported as the most common non-infectious manifestations, found in approximately 30% of all patients with CVID. Non-infectious GI diseases have also been documented in more than 15% of cases, including IBD, chronic diarrhea, GI bleeding, and diverticulosis. Malabsorption develops in 6% of individuals with CVID [7]. A similar prevalence of autoimmune complications was observed in a study of more than 2000 patients with CVID by the European Society for Immunodeficiency Registry. They also found a positive association between the co-occurrence of autoimmunity, enteropathy, granulomas, and splenomegaly, suggesting common mechanisms of immune system dysregulation [108]. Regarding GI involvement, GI complications significantly contribute to the morbidity and mortality of patients with CVID [109]. Furthermore, GI disorders and malabsorption increase the risk of mortality by more the two-fold [8]. Chronic diarrhea also significantly impairs the quality of life of patients [110]. A large study on 623 patients with CVID reported a 17% prevalence of non-infectious GI diseases, with one-third of them developing malnutrition. Enteropathy clinically mimicked IBD or CED and affected the small and large intestines in 79.4% and 50% of cases, respectively. Histological features (biopsies available in 34 participants) included villous atrophy (32.4%), nonspecific inflammation (8.8%), nodular lymphoid hyperplasia (8.8%), and intraepithelial lymphocytosis (64.7%) with the absence of plasma cells (47.1%) [7]. Notably, lymphocytic infiltration and granuloma formation, particularly involving the gut, are hallmarks of CTLA4 and LRBA deficiencies [111,112]. An increased incidence of GI nodular hyperplasia was observed in patients with APDS (26%), along with other features of lymphoproliferation, such as lymphadenopathy and splenomegaly [113]. Gastritis and gastropathy (not otherwise specified) were found in 40% of the patients with gastric disease; meanwhile, gastric metaplasia was present in 5.9% of patients. In a study of 30 patients with CVID who underwent upper and lower GI endoscopy, abnormalities were found in 83% of them. However, only 18 patients experienced GI-related symptoms. Half of the patients were HP-negative. Five patients had acute atrophic gastritis with vitamin B12 deficiency and anti-gastric parietal cell autoantibodies [114].

### 7.3. GI Malignancies

Malignancies are a significant cause of mortality in patients with PADs. Surprisingly, CVID is the second most prevalent cancer within IEI, following the syndrome associated with chromosomal instability and DNA repair defects [115]. In an Italian cohort, gastric cancer was found to be the leading cause of mortality [116]. In a meta-analysis, the overall prevalence of malignancy was 8.6%, with a predominance of lymphoma (4.1%) and gastric cancer (1.5%). The most common lymphoproliferative malignancies affecting the GIT are MALT lymphomas. Autoimmunity and malabsorption were more frequent in patients with malignancy than in those without, suggesting a role of immune dysregulation [117]. An increased risk of cancer has also been described in patients with atrophic gastritis, interstitial lung disease, arthritis, or thrombocytopenia. In several cases, monogenic causes of CVID have been identified, such as CTLA4 deficiency, NFkB1 deficiency, and APDS 1, which indicate a multifactorial pathogenesis with a significant genetic contribution. Notably, immune dysregulation and a high prevalence of non-infectious complications are hallmarks of these deficiencies [118]. Other genes associated with increased cancer susceptibility include BRCA1, RABEP1, EP300, and KDM5A [119].

### 8. Therapy of GI Complications

Therapeutically, corticosteroids and/or mesalazine may be effective as first-line therapies. However, some patients require other immunosuppressive agents (e.g., azathioprine and cyclosporine). Biological treatment has been reported in only a very limited number of patients with PAD in contrast to classical forms of IBD. The biologics indicated in these patients in particular include TNFα inhibitors (adalimumab, infliximab) that show a very low level of efficacy suggesting a different underlying pathophysiological mechanism. Other biologics such as vedolizumab or ustekinumab have been used anecdotally. Adjustment

of immunoglobulin replacement therapy with increased doses to achieve sufficient serum IgG levels should be considered [120–122]. In some cases, the identification of monogenic causes allows the initiation of disease-specific treatment, such as abatacept in CTLA4 or LRBA deficiencies [123,124], and mTOR and PI3K inhibitors in APDS [125,126]. The treatment options should also involve routine upper and lower GI endoscopies, which could be actively offered, particularly in all patients with CVID. Because a causal relationship has been described between HP infection, mucosa-associated lymphoid tissue (MALT) lymphoma, and gastric cancer, screening for HP infection (preferential urea breath test, stool antigen immunoassay, and/or endoscopic biopsy), assessment of serum vitamin B12 and iron levels, and additional tests are needed. In positive cases, ATB eradication is indicated [127]. Other nutrients important for the proper function of the immune system such as vitamin A or zinc may be missing [128]. A significant number of patients with PADs may also have a low serum concentration of vitamin D that increases the risk of osteopenia and osteoporosis [129–131]. Furthermore, reduced high-density lipoprotein (HDL) cholesterol and apo A-I, and increased levels of oxidized low-density lipoprotein (LDL) cholesterol contribute to elevated cardiovascular risk [132,133]. The assessment of nutrition status (including BMI and serum levels of particular nutrients), cardiovascular risk factors (lipid metabolism, arterial hypertension, and other cardiovascular diseases), and bone density should be considered.

## 9. Conclusions

GI complications significantly contribute to morbidity and mortality in patients with PADs. They include a broad spectrum of infectious and non-infectious complications that may mimic various diseases, such as IBD and CED, but possess different immunological features. The spectrum of infections includes *Salmonella* spp., *Campylobacter* spp., *Giardia*, noroviruses, and HP, which may contribute to an increased risk of gastric cancer and should be actively screened and eradicated in positive cases. Therefore, serological detection methods need to be avoided. GI involvement may also occur in a significant number of asymptomatic patients. Therefore, upper and lower endoscopy should be actively offered, particularly in patients with CVID, along with other additional tests, such as assessment of serum vitamin B12 and iron levels on a regular basis. Other nutrients (such as vitamin A and D, and zinc), BMI status, cardiovascular risk (including lipid metabolism and the presence of cardiovascular comorbidities), and bone density should be regularly assessed in the prevention of non-immune mediated complications. Genetic testing is routinely indicated in patients with XLA to confirm the diagnosis; however, it should be considered in patients with other PADs and severe non-infectious complications. The identification of the underlying genetic defect is crucial not only for genetic counseling but also for disease-specific management. The diagnosis and treatment of GI involvement in PADs is challenging and requires a multidisciplinary approach. Further research is needed to better understand the pathophysiology of non-infectious complications and to discover novel therapeutic and diagnostic options.

**Author Contributions:** T.M. contributed to the conceptualization, methodology, visualization, writing, editing, and funding acquisition; J.S. contributed to the writing, editing, and visualization; I.K. contributed to the conceptualization, methodology, supervision, and funding acquisition. All authors have read and agreed to the published version of the manuscript.

**Funding:** This work was supported by the grant of the Czech Health Research Council (project no. NU22-05-00402) and European Union's Horizon 2020 Research and Innovation Program (grant agreement no. 952594, ERA Chair project DRIFT-FOOD).

**Institutional Review Board Statement:** Not applicable for this work.

**Informed Consent Statement:** Not applicable for this work.

**Data Availability Statement:** Not applicable for this work.

**Conflicts of Interest:** The authors declare no conflict of interest.

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
