# Peer review of "Gastrointestinal Involvement in Primary Antibody Deficiencies"

_gastrointestdisord, doi:10.3390/gidisord5010006_

Round 1

Reviewer 1 Report

No reference is made to biological treatments, particularly in the management of intestinal inflammatory processes. Anti-TNFa or vedolizumab or ustekinumab have been used in this regard.

There is no reference to the importance of nutritional status, capital in these patients.

Reviewer 2 Report

This is a nice and comprehensive review on gastrointestinal complications of unfrequent diseases (PADs), which nonetheless should be borne in mind by specialists during the daily problem-solving practice when faced with unusual cases.

I have only minor points to suggest:

1) Introduction, line 41: granulomatous inflammation is not limited to the lung, lymph nodes are also (relatively) frequently involved;

2) line 140: class-switch should be class-switched

3) Figure 2 needs explanation of the abbreviations, as not all readers are familiar with immunology terms

4) Table 1, CVID: delete "m" from granulomatous and add a comma; change "manifestation" to "manifestations"

4) The Authors may add a table to summarize all GI complications of PADs, so as to allow the reader to take a preliminary glance at the specific conditions that will be discussed in the text.

Reviewer 3 Report

The authors present a review of the situation in the gastrointestinal tract in orimary antibody deficiencies (PADs).

The review is well organized and appropriately illustrated.

One point I would have liked to see more discussion of autoimmunity, such as the mechanism of the predilection for sIgADs, or the cause of the high incidence of autoimmune diseases (ITP, hemolytic anemia) in the blood system.

Reviewer 4 Report

This MS aims to review GI aspects of PAD

SPECIFIC COMMENTS

1. Whilst some background is reasonable and required, the MS focuses on much more than GI involvement/components: the MS should be revised and reformatted to ensure that the required focus is acheived

2. Use relevant abbreviations: Crohn disease (CD) for example

3. Please ensure to follow person first concepts throughout. The term sIgAD patients should be "patients with sIgAD" for example

4. Table 1 refers to a publication, without reference

5. Fig uses a reference also, but again without acknowledgement

6. There are a number of excessively long paragraphs - section 5 for example

7. Please review all references and correct as required to ensure that all fit the journal requirements. Reference 31 is one such example

Round 2

Reviewer 4 Report

Thank you for your revisions to date

1. The MS still contains very long paragraphs. In order to make these more digestible and enhance readability, it would be helpful to divide these long paragraphs into smaller paragraphs

2. There remain inconsistencies in the references. Some journal names are in italics and abbreviated, while others are written in full and not in italics. All should be reviewed and corrected at this stage
